# Do Wildfires Cause Changes in Soil Quality in the Short Term?

**DOI:** 10.3390/ijerph17155343

**Published:** 2020-07-24

**Authors:** Valeria Memoli, Speranza Claudia Panico, Lucia Santorufo, Rossella Barile, Gabriella Di Natale, Aldo Di Nunzio, Maria Toscanesi, Marco Trifuoggi, Anna De Marco, Giulia Maisto

**Affiliations:** 1Dipartimento di Biologia, Università degli Studi di Napoli Federico II, Via Cinthia, 80126 Napoli, Italy; valeria.memoli@unina.it (V.M.); speranzaclaudia.panico@unina.it (S.C.P.); g.maisto@unina.it (G.M.); 2Parco Nazionale del Vesuvio, Via Palazzo del Principe c/o Castello Mediceo, 80044 Ottaviano (NA), Italy; rbarile@epnv.it; 3Dipartimento di Scienze Chimiche, Università degli Studi di Napoli Federico II, Via Cinthia, 80126 Napoli, Italy; gabriella.dinatale@unina.it (G.D.N.); maria.toscanesi@unina.it (M.T.); marco.trifuoggi@unina.it (M.T.); 4CeSMA-Centro Servizi Metrologici e Tecnologici Avanzati, Università degli Studi di Napoli Federico II, Corso Nicolangelo Protopisani, 80146 San Giovanni a Teduccio (NA), Italy; aldo.dinunzio@unina.it; 5Dipartimento di Farmacia, Università degli Studi di Napoli Federico II, Via Montesano, 80131 Napoli, Italy; ademarco@unina.it

**Keywords:** sequential extraction, organic matter content, potential toxic trace elements, microbial activity, soil quality index, Vesuvius National Park

## Abstract

Wildfires have high frequency and intensity in the Mediterranean ecosystems that deeply modify the soil abiotic (i.e., pH, contents of water, organic matter and elements) and biotic properties (i.e., biomass and activity). In 2017, an intense wildfire occurred inside the Vesuvius National Park (Southern Italy), destroying approximately 50% of the existing plant cover. So, the research aimed to evaluate the fire effects on soil quality through single soil abiotic and biotic indicators and through an integrated index (SQI). To achieve the aim, soil samples were collected inside the Vesuvius National Park at 12 sampling field points before fire (BF) and after fire (AF). The findings highlighted that in AF soil, the contents of water and total carbon, element availability, respiration and the dehydrogenase activity were lower than in BF soil; in contrast, pH and hydrolase activity were significantly higher in AF soil. The microbial biomass and activity were affected by Al, Cr and Cu availability in both BF and AF soils. Despite the variations in each investigated soil abiotic and biotic property that occurred in AF soil, the overall soil quality did not significantly differ as compared to that calculated for the BF soil. The findings provide a contribution to the baseline definition of the properties and quality of burnt soil and highlight the short-term effects of fire on volcanic soil in the Mediterranean area.

## 1. Introduction

In Mediterranean ecosystems, fires have high frequency and intensity due to dry and warm climatic conditions and to the presence of vegetation prone to widespread crown fires [1,2]. In these ecosystems, fires act as evolutionary pressure, shaping plant traits and stimulating adaptive responses by forests [3], but they also play a key role in modifying soil abiotic properties, microorganism abundance and activity and, in turn, the relationships between plant and soil communities [4]. This is particularly noticeable in arid zones where biological soil activity is dependent on climate [5].

The main direct effect of fires on soils is heating, whereas, the indirect ones are linked to ash-bed effects, vegetation recovery, topography and management. After fires, ash formation and deposition on soil lead to pH increase, because of the release of alkaline elements [6]. As ashes can be hydrophobic, soil water repellence increases whereas water permeability decreases [7]. In addition, the element availability also increases due to organic matter destruction [8]. The increased availability of nutrients and metals can lead to a loss of nutrients by erosion and leaching processes [9] and to a remobilization of potential toxic metals [10]. In fact, in burnt soils, Stankov-Jovanovic et al. [11] found an increased content of heavy metals, particularly in the fraction available to organisms. The post-fire increased metal availability can alter soil organism activities and, in turn, important soil processes such as the recycling of organic matter, the element biogeochemical cycles and plant productivity [12].

In Mediterranean ecosystems, the recurrence of fires four times in less than 50 years can endanger soil quality, affecting the nutrient cycle and microbial activity [13]. Many studies have analyzed soil functionality after fire events [4,14,15,16], but few studies assessed the effects of the whole soil quality through single and integrated indices. Soil quality is related to chemical composition, biological biomass and activity, such as respiration and enzymatic activities and fungi/bacteria ratio [5,13,17,18,19,20]. All of these properties are fundamental tools to enhance the resilience of burnt ecosystems [21] and are crucial for healthy plant growth and for improving soil quality [14,22].

In 2017, the extensive and intensive fires, occurring inside the Vesuvius National Park, destroyed approximately 50% of the existing plant cover, reducing the biodiversity and the productivity of the whole ecosystems. So, the aim of the research was to evaluate the fire effects on soil quality through single soil abiotic and biotic indicators (i.e., organic matter and water contents, pH, element availability, respiration, C_fung_/C_mic_ and enzymatic activities) and through an integrated index. To achieve the aim, soil chemical and biological analyses were performed one year after the fire occurrence (2018) and compared to those performed on soil sampled one year before it (2016). As changes in abiotic and biotic soil properties were expected, the hypothesis behind the research was an overall worsening of soil quality after fire. To verify the hypothesis, soil quality, expressed as soil quality index (SQI), was calculated according to Leitgib et al. [23] in before fire (BF) and after fire (AF) soils and compared.

## 2. Materials and Methods

### 2.1. Study Area

The Vesuvius National Park, established in 1995 and located 12 km SE of Naples, covers an area of 8482 ha and contains Mt. Somma (geographic coordinates: 40°50′14” N–14°25′41” E; maximum height: 1132 m a.s.l.), the original volcano, and Mt. Vesuvius (coordinates: 40°49′17” N–14°25′32” E; maximum height: 1281 m a.s.l.), which originated from the 79 A.D. eruption. At present, Vesuvius is in a quiescent phase and is covered by native Mediterranean vegetation based in trees (such as *Quercus ilex* L., *Acer opalus* L. and *Ulmus minor* L.) and shrubs (such as *Mirtus communis* L., *Laurus nobilis* L., *Viburnum*
*tinus* L., *Cistus* sp. *Ginesta* sp.), but specimens of *Pinus nigra* L. and *Robinia pseudoacacia* L. are also present [24,25]. The Vesuvius mountain is characterized by Mediterranean climatic conditions with dry summers and rainy autumns and winters (mean annual temperature: 13.2 °C; annual precipitation: 960 mm, Osservatorio Vesuviano). The soils of the Vesuvius National Park show a silty-clay texture [26] and are classified as Lepti-Vitric Andosols [27].

In summer 2017, a wildfire occurred, causing the loss of more than 50% of the existing plant cover (www.forbes.com; www.earth.esa.int) and it was suppressed by seawater.

### 2.2. Soil Sampling

The soil samplings were performed in fall 2016 (BF) and fall 2018 (AF), respectively, one year before and after the wildfire, occurred in summer 2017. The average temperature and the rainy days measured in the month before samplings, were 18.5 °C and 18 rainy days/month for 2016, and 19.9 °C and 15 rainy days/month for 2018. Meteorological data from September 2016 to September 2018 are reported in Figure A1. After fire, at 12 field points (Figure 1) inside a high intensity (level 4) burnt area [28], five subsamples of surface soil (0–10 cm) were collected, after litter removal, and mixed together in order to obtain a homogeneous sample. Before sampling, the litter was removed in order to study only the surface soil system, where the biological activity is higher [29], and also to allow the comparison with the results of the campaign of 2016. The field points were selected according to those investigated in the sampling campaign of fall 2016. To reduce climatic variability, soil samplings occurred after 15 days without rain. The samples were collected in glass containers and sent to the laboratory in an ice box, and then stored at 4 °C.

### 2.3. Soil Chemical Analyses

In the laboratory, the soil samples were sieved (<2 mm) and analyzed for pH, water (WC) and organic matter (OM) contents as well as for the total C and N contents. Soil pH was measured with a pH-meter on aqueous extract obtained by adding distilled water to soil (2.5:1 = *w*:*w*). Water content was determined gravimetrically by drying fresh soil at 105 °C until a constant weight was reached. Organic matter content was calculated by multiplying the C_org_ by 1.724 [30] that was measured by gas-chromatography (Thermo Finnigan, CNS Analyzer) on dried and pulverized samples (5 mg d.w.), previously saturated with HCl (10% *v*:*v*). Total carbon and nitrogen concentrations were determined by gas-chromatography (Thermo Finnigan, CNS Analyzer) on dried and pulverized samples.

The exchangeable (F1), reducible (F2), oxidizable (F3) and residual (F4) fractions of Al, Ca, Cr, Cu, Fe, K, Mg, Mn, Pb and V were evaluated applying the sequential extraction as suggested by the Bureau of Community Research (BCR) [31]. Samples were analyzed by ICP-MS (Aurora M90 Bruker, Billerica, MA, US). Details of the extraction methods are reported in [32]. The sum of the concentrations of each element in the four fractions is considered as pseudo-total [32].

### 2.4. Biological Analyses

The biological analyses (microbial and fungal carbon, respiration, hydrolase and dehydrogenase activities) were performed on soil samples stored at 4 °C within three days of the soil sampling. Briefly, the microbial carbon (C_mic_) was evaluated according to Anderson and Domsch [33], whereas the percentage of microbial C present as fungal C (C_fung_) was obtained by converting total fungal biomass (TFB) to fungal C content on the basis of mean fungal values of C/N ratio [34] and N content [35]. The total fungal biomass was evaluated after staining with aniline blue, by the membrane filter technique [36], determining hypha length by the intersection method [37] with an optical microscope (Optika, B-252).

Respiration was estimated as CO_2_ evolution from the samples at 55% of water holding capacity after incubation in tight containers for 10 days at 25 °C by NaOH absorption followed by two-phase titration with HCl [38].

Hydrolase (HA) and dehydrogenase (DHA) activities were evaluated using fluorescein diacetate (1 mg mL^−1^) and 2,3,5-triphenyltetrazolium chloride 1.5%, respectively, as substrates according to Memoli et al. [39]. The results were, respectively, expressed as mmol of fluorescein (FDA) and triphenylformazan (TPF) produced in 1 min for 1 g of dried soil.

### 2.5. Soil Quality Index (SQI)

An integrated soil quality index was calculated taking into account the physico-chemical and biological parameters that were ranked from 0 to 1, respectively, reflecting low and high quality, according to Leitgib et al. [23]. The scores were assigned applying the more is better or less is better functions. The more is better function was applied to organic matter and water contents, C and N concentrations [23], macro-nutrient availability (i.e., sum of F1, F2 and F3 concentrations of Ca, K, Mg and Mn), C_fung_/C_mic_, Resp HA and DHA, for their roles in soil fertility, water partitioning and nutrient availability. On the contrary, the less is better function was applied to micro-nutrient and trace metal availability (i.e., sum of F1, F2 and F3 concentrations of Al, Cr, Cu, Pb and V) because their high concentration is potentially toxic for soil organisms, according to Marzaioli et al. [40].

For each site and each sampling time, the SQI was calculated, summing the parameter scores and dividing for the number of parameters, as reported by Andrews et al. [41]:$$\mathrm{SQI}=\overset{n}{\underset{i=1}{\sum }}\frac{Si}{n}$$
where SQI is soil quality index, S is the score assigned to each parameter and n is the number of the investigated parameters.

All of the soil parameters measured were used in the standardized principal component analysis, PCA [42] to highlight those involved as main factors influencing BF and AF soil qualities. As the principal components (PCs) with eigenvalues <1 have less variation than the individual variable, only the PCs with eigenvalues >1 were considered for the identification of the minimum data set (MDS) [43]. Within each PC, parameters with absolute values within 10% of the highest weighted loading were selected for the MDS. When, for each PC, more parameters showed the values to be selected for the MDS, the parameters that were not correlated were performed for the selection [44].

### 2.6. Statistical Analyses

The normality of the distribution of the data sets was assessed by the Shapiro–Wilk test.

The paired *t*-tests or the signed rank tests according to normal or non-normal distribution of the data, respectively, were performed to evaluate the differences in each investigated parameter between BF and AF soils.

The ANOVA test was performed to compare the differences among the different element fractions (F1, F2, F3 and F4) within both BF and AF soils.

In order to predict if the biological variables were related to the same soil abiotic properties (pH, water and organic matter content, total C and N contents and the F1, F2 and F3 of the elements) in BF and AF soils, multiple linear regressions were carried out to before and after fire data sets, separately. The linearity of the data, the independence, the homogeneity and the normality of residuals were tested and adjusted, if required, before the multiple linear regressions’ performance.

The statistical tests were considered significant when *p* < 0.05.

The *t*-tests, ANOVA and multiple regressions were performed by the Systat_SigmaPlot_12.2 software (Jandel Scientific, San Rafael, CA, USA). The graphs were created by SigmaPlot12 software (Jandel Scientific, San Rafael, CA, USA).

## 3. Results

The mean values of organic matter and water contents, as well as C and N contents, were higher in BF than AF soil with significant differences for C, organic matter and water contents (Table 1). By contrast, the mean value of pH was significantly higher in AF soil (Table 1).

The comparison of the element contents in each fraction between BF and AF soils highlighted that Ca, Cr, K and Mg in the F1, Mn in the F2, Al, Ca, Cr, Cu, K and Mn in the F3 and V in the F4 were significantly higher in BF than AF soil (Table 2); instead, Al in the F1, Cr and Cu in the F2, V in the F3 and Ca, K and Pb in the F4 were significantly higher in AF than BF soil (Table 2). The pseudo-total concentrations of the investigated elements slightly varied between BF and AF soils, with the exception of K and Pb that were significantly higher in AF soils (Table 2). All the investigated elements, with the exception of Pb and Cu in BF soil, were significantly higher in the F4 as compared to the other fractions in both BF and AF soils (Table 2). In both BF and AF soils, the F1 frequently showed the lowest values, particularly for Cr, Pb and V, as compared to F2 and F3. The availability of K and Mg in AF soil did not significantly vary among F1, F2 and F3 (Table 2).

The C_fung_ (BF soils: 0.02 ± 0.01 mg g^−1^ C d.w.; AF soils: 0.010 ± 0.003 mg g^−1^ C d.w.) and C_mic_ (BF soils: 1.43 ± 0.43 mg g^−1^ C-CO_2_ d.w.; 1.75 ± 0.47 mg g^−1^ C-CO_2_ d.w.) values did not significantly vary between BF and AF soils. The C_fung_/C_mic_ ratios did not statistically vary between BF and AF soils (Figure 2). The basal respiration and the dehydrogenase activity were significantly higher in BF than AF soil (Figure 2). In contrast, the hydrolase activity was significantly higher in AF than BF soil (Figure 2).

The multiple regressions highlighted that soil biological properties are impacted more by available fractions of elements than by soil pH, water content, total C and N contents and organic matter content. In particular, the F1, F2 and F3 fractions of Al, Cr and Cu affected all the investigated soil biological properties in both BF and AF soils (Table 3).

The values of SQI amounted approximately to 0.4 in both BF and AF soils and did not significantly vary between the burnt and unburnt soils (Figure 3). Instead, the properties that mainly affected the BF soil quality were WC, Resp, HA, and availability (i.e., sum of F1, F2 and F3 contents) of Cu, Fe, K, Mg, Mn and V; whereas, those for AF soil quality were WC, Resp, C_fung_/C_mic_ and availability of Ca.

## 4. Discussion

In the investigated area, wildfire events modified properties and nutrient and trace element concentrations in soil. The lower total C content detected in AF soil as compared to the BF soil, suggested that wildfires enhance the C losses from soil [45], burning the C labile fractions [46]. Instead, the remaining C could be less inflammable or protected from volatilization, by its incorporation into the soil inorganic compounds [47]. Despite the significant C decrease, changes in N concentrations between AF and BF soils were not noticeable, according to several studies that did not find considerable decrease in soil N pools [48]. This can probably be linked to site organic matter quality that forms recalcitrant N compounds, remaining in soils [16]. Water repellency—a soil property that can occur under natural conditions [49,50], but that can be enhanced by forest fires in Mediterranean environments [13,51]—could be mostly responsible for the significant decrease of water content observed between BF and AF soils in the investigated area. Other factors can also have an important influence on this response, such as the lack of organic soil cover, and the scarcity of plant cover, thus favoring evaporation from the exposed soil and counteracting the effect of limited vegetation regrowth in soil water content.

Instead, ash accumulation and cation release from organic matter and vegetation burned could contribute to the significant increase of pH between BF and AF soil in the investigated area [8].

Taking into account both the pseudo-total contents and available fractions, it can be supposed that wildfires generated wide variations in the soil chemical composition with different trends depending on each element. In particular, the increase in AF soil of the K, Mn and Pb pseudo-total contents, even though Mn was not significant, suggests their inputs deriving by the burnt biomass [52,53]. An overall evaluation highlighted a decrease of the element (especially, for macro- and micro-nutrients) availability due to the wildfires, particularly for the oxidizable fraction (F3). These findings disagree with studies about prescribed fires or wildfires of low intensity where a general increase of element availabilities occurred [54,55], but they agree with studies on intense prescribed fires where a decrease of element availability occurred after one year since fires [56,57,58,59]. The exchangeable (F1) and the oxidizable (F3) fractions are more impacted by fire, than F2 and F4, because F1 is composed of element forms which are very easy to wash off, and thus transferred through rainfall to other parts of the environment; whereas F3 is the part of element linked to OM and it is mobilized during the combustion [11]. In the investigated area, rainfall events and erosion have probably contributed to the decrease of F1 and F3 element contents [60]. However, in both BF and AF soils, the lowest values of the elements in the exchangeable (F1) fraction also suggests their scarce water solubility, due to the linkages with volcanic minerals [61]. The particular low presence of potential toxic elements (i.e., Cr, Pb and V) in F1 could be due to the scarce potentiality of volcanic rocks to bind them [32]. On the other hand, the reducible (F2) and residual (F4) fraction contents were less affected by fire in the investigated soil. In fact, only the F2 of Cr, Cu and Mn and only the F4 of K and P increased in AF soil. The weak impact of fire on F2 and F4 can probably happen because the reducible fraction is less mobilized by oxidative processes, whereas the residual fraction is an inert phase of element, which cannot be easily mobilized [11]. The high amount in F4 in both BF and AF soils, suggests that the investigated elements are strongly linked to soil solid phase, as previously detected in other monitoring campaigns inside the Vesuvius National Park [32].

The alterations produced by fire on soil chemical properties (i.e., organic matter and water contents, pH and element availability) modified the habitat, causing important responses by the microbial community [62]. In the investigated area, the lack of significant differences in C_fung_/C_mic_ between BF and AF soil could suggest the high capability of microorganisms to recover [63] or, conversely, their acquisition of fire-resistant structures that allow withstanding the adverse properties of burnt soils [64]. In addition, the observed lack of difference could also be due to the low values of C_fung_/C_mic_, due to the low proportion of fungal biomass observed in the investigated soil, also before fire. The low fungal biomass can be due to local soil conditions, such as alkaline pH and the low C/N ratio (approximately 12) that favor bacteria instead of fungi [65]. Despite no variations in total biomass a shift of microbial community with different functional groups and metabolic properties could be supposed [66]. The hypothesis, that in the investigated area changes in different functional groups of microorganisms occurred, is corroborated by significant differences in some fundamental ecological processes, such as a conspicuous reduction of respiration and DHA activity in AF soil that contribute to decomposition and nutrient cycles [67]. The lower respiration, after wildfire occurrence, likely can be due to the small amount of litter and to the main carbonaceous material, which can limit the microbial decomposition and respiration [68]. However, in the investigated burnt soil, a negative effect on respiration and DHA activity due to the raised Al, Cr and Cu availabilities (significantly higher in AF than BF soil, respectively, in F1, F2 and F3), cannot be excluded, although the role of metal availability already existed before the wildfires. This supposition is corroborated by the multiple regressions. Instead, according to Fuentes-Ramirez et al. [69] one year after a severe fire in forests of south-central Chile, HA activity appeared to be stimulated by soil Al, Cr and Cu availabilities. The different behaviors observed for HA and DHA activities could be explained as DHA, an intracellular enzyme, and could be less affected by the properties of the surrounding soils, compared to HA, an extracellular enzyme, as demonstrated for other extracellular enzymes [70].

Although the comparison of the soil properties in BF and AF displayed noticeable changes in each considered soil property (i.e., element availability and soil microbial activity), the SQI did not show appreciable changes in the whole quality due to wildfire occurrence. This evidence could be due to the fact that only WC and Resp affected the quality of both BF and AF soils. In addition, the BF soil quality was affected by high Cu, Fe, K, Mg, Mn and low V availability; instead, the AF soil quality was also affected by the slight increase of bacterial C and by the decrease of Ca availability. The different or opposite responses to fire of the investigated soil properties did not cause changes in the overall soil quality between BF and AF soils. These findings suggest that changes in a single soil property can be compensated when their interactions are considered [38].

## 5. Conclusions

One year after fire, substantial variations in each investigated soil chemical property occurred, affecting the soil microbial biomass and activity. In particular, the decrease of organic matter, C and water contents and element availability as well as the increase in pH, caused a decrease of microbial respiration and DHA activity and an increase of HA activity.

Despite the variations in each investigated soil abiotic and biotic property that occurred in AF soils, the overall soil quality did not significantly differ as compared to that calculated for the BF soil (i.e., the SQIs were approximately 0.4 in both BF and AF soils). The lack of meaningful differences in the overall soil quality was due to the fact that some abiotic and biotic properties increased whereas others decreased after fire occurrence.

Therefore, the research highlighted that single indicators were not representative of the overall soil quality in the investigated volcanic Mediterranean area.

Nevertheless, the obtained results highlight the fire effects on soil quality for the brief term and further investigations will be performed in order to assess its effects in the long term.

## Figures and Tables

**Figure 1 ijerph-17-05343-f001:**
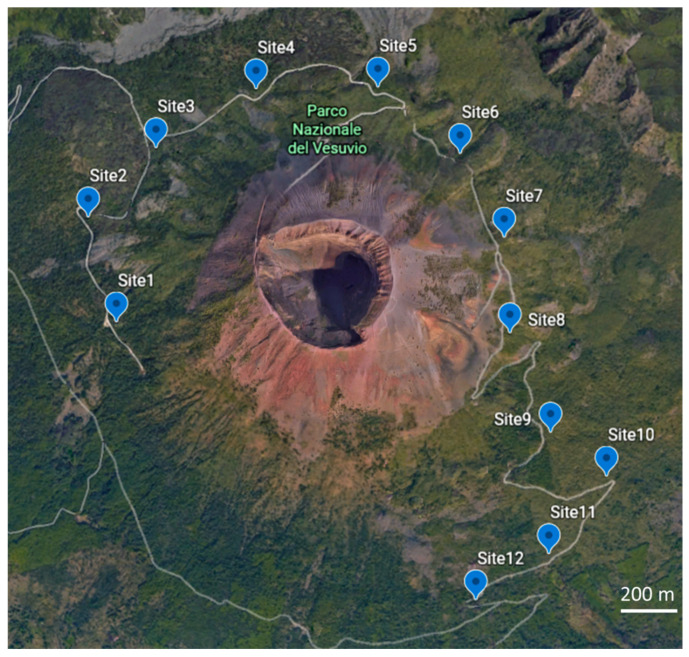
Map of the sampling field points (blue points) inside the Vesuvius National Park.

**Figure 2 ijerph-17-05343-f002:**
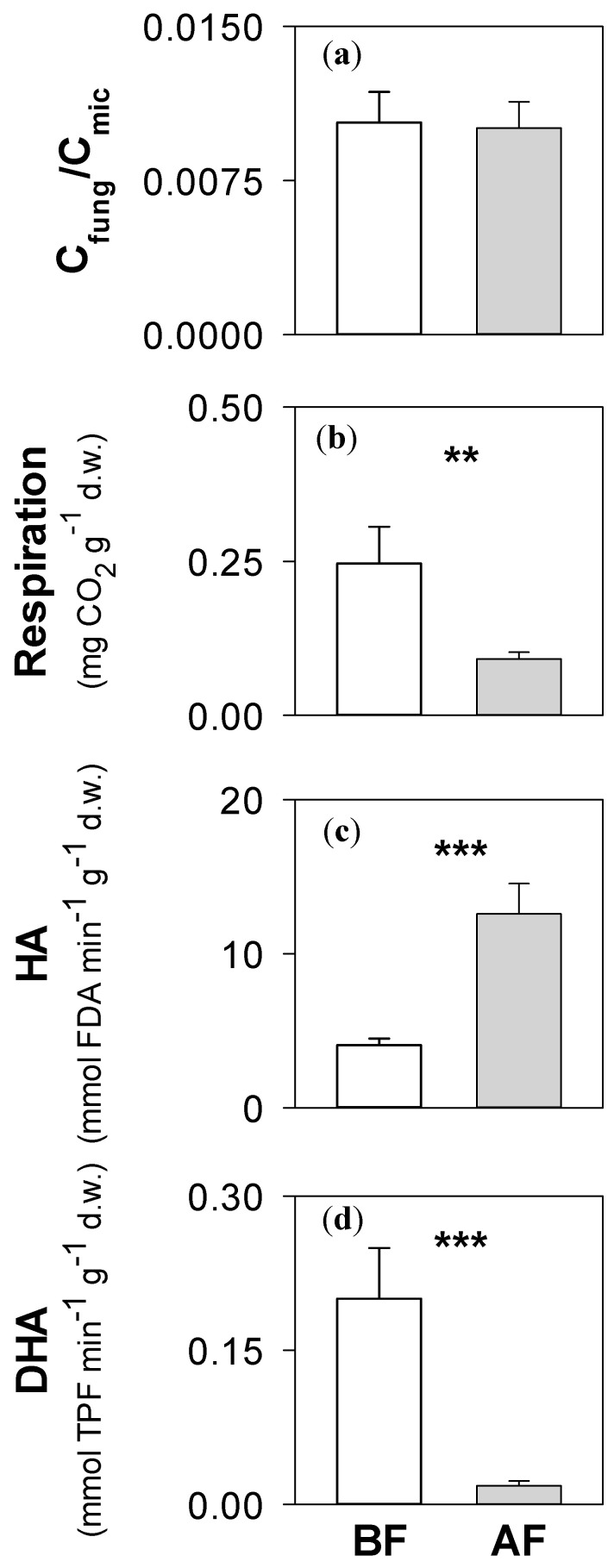
Mean values (± e.s.) of (**a**) C_fung_/C_mic_, (**b**) microbial respiration, (**c**) hydrolase activity (HA), (**d**) dehydrogenase activity (DHA) measured before (BF) and after fire (AF) in soils collected on the Vesuvius National Park, Southern Italy. The asterisks indicate significant differences (*t*-test—*: *p* < 0.05, **: *p* < 0.01, ***: *p* < 0.001) in soil properties between BF and AF soils.

**Figure 3 ijerph-17-05343-f003:**
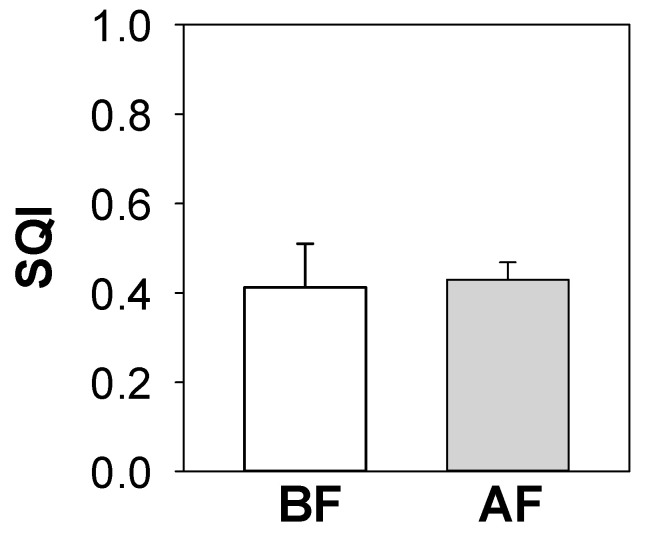
Mean values (± e.s.) of soil quality index (SQI) based on soil chemical and biological properties calculated before (BF) and after fire (AF) in soils collected on the Vesuvius National Park, Southern Italy.

**Table 1 ijerph-17-05343-t001:** Mean values (± s.e.) of organic matter content (OM), total contents of carbon (C), nitrogen (N), and water (WC), and pH in soils collected inside the Vesuvius National Park before (BF) and after (AF) fire occurrence. The asterisks indicate significant differences (*t*-test, at least *p* < 0.05) of each investigated soil property between BF and AF.

	OM (% d.w.)	C (% d.w.)	N (% d.w.)	WC (% d.w.)	pH
**BF**	7.53 *	5.46 *	0.43	45.2 *	7.11 *
	(±0.03)	(±0.13)	(±0.13)	(±19.1)	(±0.20)
**AF**	4.82	2.89	0.23	17.0	7.42
	(±0.49)	(±0.60)	(±0.05)	(±2.30)	(±0.11)

**Table 2 ijerph-17-05343-t002:** Mean values (± s.e.) of the exchangeable (F1), reducible (F2), oxidizable (F3) and residual (F4) fractions of Al, Ca, Cr, Cu, Fe, K, Mg, Mn, Pb and V (µg g^−1^ d.w.) in soils collected inside the Vesuvius National Park before (BF) and after (AF) fire occurrence. The sum of the contents of each element in the four fractions is reported as pseudo-total. The asterisks indicate significant differences (*t*-test, at least *p* < 0.05) of the element in each fraction and pseudo-total contents between BF and AF. Different uppercase letters indicate significant differences (ANOVA, *p* < 0.05) among the fractions (F1, F2, F3 and F4) for each element.

Element	Soil	F1	F2	F3	F4	Pseudo-Total
**Al**	BF	401 * B	783 AB	3901 * AB	11968 A	17053
		(±78.4)	(±206)	(±917)	(±1457)	(±1681)
	AF	612 B	916 AB	1739 AB	1,405 A	15671
		(±115)	(±75.0)	(±244)	(±2500)	(±2643)
**Ca**	BF	2708 *AB	1241 B	2122 * AB	6161 *A	1,232
		(±547)	(±411)	(±40.9)	(±394)	(±616)
	AF	1619 AB	961 B	1361 AB	8634 A	12575
		(±266)	(±119)	(±250)	(±1598)	(±1858)
**Cr**	BF	0.02 * B	0.00 * B	0.65 * AB	1.16 A	1.83
		(±0.01)	(±0.00)	(±0.35)	(±0.19)	(±0.52)
	AF	0.01 B	0.11 AB	0.16 AB	0.91 A	1.17
		(±0.00)	(±0.01)	(±0.01)	(±0.24)	(±0.24)
**Cu**	BF	0.98 B	0.28* B	17.6 * A	12.1 A	30.93
		(±0.40)	(±0.14)	(±2.61)	(±1.04)	(±2.51)
	AF	1.44 B	0.68 B	3.90 B	27.6 A	33.62
		(±0.25)	(±0.06)	(±0.43)	(±12.1)	(±12.1)
**Fe**	BF	91.0 C	594 BC	1290 AB	7509 A	9483
		(±41.0)	(±192)	(±516)	(±753)	(±634)
	AF	145 C	609 BC	927 AB	7268 A	8494
		(±97.4)	(±100)	(±188)	(±1211)	(±1063)
**K**	BF	387 * BC	146 C	808 * B	10843 * A	12184 *
		(±50.8)	(±27.7)	(±181)	(±1089)	(±1027)
	AF	242 B	150 B	380 B	14,211 A	14982
		(±27.9)	(±18.4)	(±55.3)	(±1707)	(±1713)
**Mg**	BF	420 * A	97.5 B	441 A	2567 A	3525
		(±137)	(±36.3)	(±161)	(±222)	(±364)
	AF	208 B	129 B	256 B	2576 A	3169
		(±8.77)	(±22.5)	(±42.1)	(±378)	(±402)
**Mn**	BF	59.3 A	47.1* A	30.9 * A	147 A	284
		(±27.7)	(±24.0)	(±7.65)	(±14.0)	(±55.2)
	AF	54.1 AB	20.5 B	12.5 B	233 A	320
		(±17.0)	(±4.20)	(±3.47)	(±114)	(±115)
**Pb**	BF	0.47 B	4.61 AB	14.9 A	3.65 * AB	23.6 *
		(±0.13)	(±1.69)	(±7.31)	(±0.40)	(±8.95)
	AF	0.53 B	4.70 AB	12.9 AB	66.7 A	84.8
		(±0.05)	(±0.94)	(±2.74)	(±6.15)	(±12.5)
**V**	BF	0.28 C	1.37 BC	9.67 * B	25.2 * A	36.5
		(±0.08)	(±0.17)	(±1.97)	(±2.63)	(±1.99)
	AF	0.34 B	1.45 B	20.4 A	15.3 A	37.49
		(±0.13)	(±0.12)	(±2.20)	(±3.49)	(±2.41)

**Table 3 ijerph-17-05343-t003:** Multiple regressions (*p* < 0.05) among soil pH, water content (WC), organic matter content (OM), total content of C and N and available fractions (F1, F2, F3) of element contents and biotic variables measured before (BF) and after fires (AF) in soils collected inside the Vesuvius National Park.

BF	AF
**F1**	
C_fung_/C_mic_ = 0.0238 − (0.0000247 Al) + (0.0445 Cr) − (0.00458 Cu)	C_fung_/C_mic_ = 0.0285 − (0.000000647 Al) − (3.068 Cr) − (0.000162 Cu)
Resp = 0.515 − (0.000533 Al) + (10.161 Cr) − (0.275 Cu)	Resp = 1.078 + (0.000206 Al) − (99.745 Cr) − (0.369 Cu)
HA = 5.203 − (0.00264 Al) + (91.802 Cr) − (2.063 Cu)	HA =−130.675 − (0.0407 Al) + (15667.992 Cr) + (53.415 Cu)
DHA = 0.583 − (0.000791 Al) + (3.989 Cr) − (0.153 Cu)	DHA = 0.267 − (0.00000946 Al) − (21.625 Cr) − (0.0814 Cu)
**F2**	
C_fung_/C_mic_ = 0.173 − (0.000293 Al) + (20.621 Cr) + (0.0107 Cu)	C_fung_/C_mic_ = 0.0217 + (0.0588 Cr) − (0.00832 Cu) − (0.00259 Pb)
Resp = −0.0883 + (0.000438 Al) + (6.453 Cr) − (0.100 Cu)	Resp = 0.346 − (0.000167 Al) − (3.448 Cr) + (0.389 Cu)
HA = −14.555 + (0.0318 Al) − (1890.699 Cr) − (1.690 Cu)	HA = −15.093 + (0.0333 Al) + (579.982 Cr) − (94.778 Cu)
DHA = 4.527 − (0.00788 Al) + (564.029 Cr) + (0.393 Cu)	DHA = 0.152 − (0.0000438 Al) − (0.470 Cr) − (0.0656 Cu)
**F3**	
C_fung_/C_mic_ = 0.00929 + (0.00000292 Al) + (0.00164 Cr) − (0.000650 Cu)	C_fung_/C_mic_ = 0.0639 − (0.0000215 Al) + (0.323 Cr) − (0.0171 Cu)
Resp = 0.350 −(0.000180 Al) + (0.762 Cr) + (0.00601 Cu)	Resp = 0.875 − (0.000199 Al) − (1.006 Cr) − (0.0719 Cu)
HA = 4.808 − (0.00188 Al) + (6.711 Cr) + (0.128 Cu)	HA = −139.128 + (0.0436 Al) − (194.563 Cr) + (27.220 Cu)
DHA = 0.0453 + (0.0000759 Al) + (0.0989 Cr) − (0.0117 Cu)	DHA = 0.103 − (0.0000118 Al) − (1.285 Cr) + (0.0347 Cu)
**pH, WC, C, N, OM and sum of the element available fractions**	
C_fung_/C_mic_ = −0.0133 + (0.0000106 Al) − (0.0216 Cr) − (0.000852 Cu)	C_fung_/C_mic_ = −0.0620 + (0.0000123 Al) − (0.254 Cr) + (0.0165 Cu)
Resp = 1.349 − (0.000503 Al) + (1.725 Cr) + (0.0157 Cu)	Resp = −0.0823 + (0.0000740 Al) − (3.125 Cr) + (0.127 Cu)
HA = 15.968 − (0.00553 Al) + (17.623 Cr) + (0.232 Cu)	HA = 113.533 − (0.0249 Al) + (663.068 Cr) − (32.653 Cu)
DHA = −0.490 + (0.000260 Al) − (0.454 Cr) − (0.0173 Cu)	DHA = 0.217 − (0.0000302 Al) − (0.128 Cr) − (0.0110 Cu)

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
