# Peer review of "Do Wildfires Cause Changes in Soil Quality in the Short Term?"

_ijerph, 2020, doi:10.3390/ijerph17155343_

Round 1

Reviewer 1 Report

In their manuscript, the authors approached the study of wildfire impact on several soil variables, also considering a synthetic soil quality index. The study presents a topic which fits the scope of the journal and is of potential interest to the IJERPH audience.

In general, the manuscript is well organized and kept focused on the topic. The full length of the paper is adequate, the statistical analysis supports the main findings and the discussion is critically organized. Then I can recommend it for publication with minor revision.

My first concern deals with the sampling strategy, which is not adequately described. What is meant for twelve sites? How far from each other were they? Did you consider them as field replicates? If you have 12 field (true) samples, then you do not need to run your lab analysis in triplicate. Please clarify.

Secondly. Please inform whether the wildfire was extinguished by using freshwater, marine water or what else (foam, extinguishing powders, etc).

Table 1. OM and organic C appear not related to each other: in BF OM < Corg (??), while in AF C should be 2.79 instead of 2.89. Please, explain.

Finally, in their research the authors found that fungal C represents less than 1% of microbial C, which is quite unexpected. Any comment on that? Is the fungal biomass determination method adeguate?

Author Response

Reviewer 1

In their manuscript, the authors approached the study of wildfire impact on several soil variables, also considering a synthetic soil quality index. The study presents a topic which fits the scope of the journal and is of potential interest to the IJERPH audience. In general, the manuscript is well organized and kept focused on the topic. The full length of the paper is adequate, the statistical analysis supports the main findings and the discussion is critically organized. Then I can recommend it for publication with minor revision.

The authors thank the Reviewer for the appreciation of the research. They also thank the Reviewer for the suggestions and corrections that improved the manuscript.

My first concern deals with the sampling strategy, which is not adequately described. What is meant for twelve sites? How far from each other were they? Did you consider them as field replicates? If you have 12 field (true) samples, then you do not need to run your lab analysis in triplicate. Please clarify.

The authors have modified the paragraph about the sampling strategy, specifying the soil type, number of sites and replicates. In addition, they have added a figure for the localization of the sampling field points and a spatial scale useful to identify the distance among them (Fig. 1). L. 100-103

Secondly. Please inform whether the wildfire was extinguished by using freshwater, marine water or what else (foam, extinguishing powders, etc).

The authors have added this information in the new version of the manuscript. L. 92-93

Table 1. OM and organic C appear not related to each other: in BF OM < Corg (??), while in AF C should be 2.79 instead of 2.89. Please, explain.

In Table 1, the authors have reported (already in the previous version of the manuscript), the total C content. Therefore, OM and C are not directly related.

Finally, in their research the authors found that fungal C represents less than 1% of microbial C, which is quite unexpected. Any comment on that? Is the fungal biomass determination method adeguate?

The authors can ensure that the determination method for the fungi is adequate, and used in many published researches.

The low fungal biomass can be due to local soil/environmental conditions. In fact, the alkaline pH and the low C/N (approximately equal to 12) make fungi less competitors than bacteria inside the microbial community (Malik et al 2016, Frontiers in Microbiol, doi: 10.3389/fmicb.2016.01247).

The authors thank the reviewer for this pertinent comment and have added this information in the new version of the manuscript. L. 296-299

Reviewer 1

In their manuscript, the authors approached the study of wildfire impact on several soil variables, also considering a synthetic soil quality index. The study presents a topic which fits the scope of the journal and is of potential interest to the IJERPH audience. In general, the manuscript is well organized and kept focused on the topic. The full length of the paper is adequate, the statistical analysis supports the main findings and the discussion is critically organized. Then I can recommend it for publication with minor revision.

The authors thank the Reviewer for the appreciation of the research. They also thank the Reviewer for the suggestions and corrections that improved the manuscript.

My first concern deals with the sampling strategy, which is not adequately described. What is meant for twelve sites? How far from each other were they? Did you consider them as field replicates? If you have 12 field (true) samples, then you do not need to run your lab analysis in triplicate. Please clarify.

The authors have modified the paragraph about the sampling strategy, specifying the soil type, number of sites and replicates. In addition, they have added a figure for the localization of the sampling field points and a spatial scale useful to identify the distance among them (Fig. 1). L. 100-103

Secondly. Please inform whether the wildfire was extinguished by using freshwater, marine water or what else (foam, extinguishing powders, etc).

The authors have added this information in the new version of the manuscript. L. 92-93

Table 1. OM and organic C appear not related to each other: in BF OM < Corg (??), while in AF C should be 2.79 instead of 2.89. Please, explain.

In Table 1, the authors have reported (already in the previous version of the manuscript), the total C content. Therefore, OM and C are not directly related.

Finally, in their research the authors found that fungal C represents less than 1% of microbial C, which is quite unexpected. Any comment on that? Is the fungal biomass determination method adeguate?

The authors can ensure that the determination method for the fungi is adequate, and used in many published researches.

The low fungal biomass can be due to local soil/environmental conditions. In fact, the alkaline pH and the low C/N (approximately equal to 12) make fungi less competitors than bacteria inside the microbial community (Malik et al 2016, Frontiers in Microbiol, doi: 10.3389/fmicb.2016.01247).

The authors thank the reviewer for this pertinent comment and have added this information in the new version of the manuscript. L. 296-299

Reviewer 2 Report

The manuscript “Do Wildfires Cause Changes in Soil Quality in Short Term? ” (No. 858218.)by Valeria Memoli described on the comparison of soil properties at the burnt sites, AF, with those at the same sites one year before the wildfires, BF. The findings provide a contribution to the baseline definition of the burnt soil properties inside the Mediterranean area, concerning volcanic soils and highlight the short-term effects of wildfires. However, also have one problem need to be improved.

  1. Abstract: The first place can not indicated the abbreviation AF and BF, should be showed full name.

    2.the line 67-71, I suggest that rewrite the aim. Please reference some published papers.

  1. I don’t know the meaning of the a b c A B C in the table1, 2. Such as 401 b B. Please show the meaning.
  2. the conclusion should add some future works for this study.
  3. please check the references format, carefully.

Author Response

Reviewer 2

The manuscript “Do Wildfires Cause Changes in Soil Quality in Short Term? ” (No. 858218.) by Valeria Memoli described on the comparison of soil properties at the burnt sites, AF, with those at the same sites one year before the wildfires, BF. The findings provide a contribution to the baseline definition of the burnt soil properties inside the Mediterranean area, concerning volcanic soils and highlight the short-term effects of wildfires. However, also have one problem need to be improved.

The authors thank the Reviewer for the appreciation of the research. They also thank the Reviewer for the suggestions and corrections that improved the manuscript.

Abstract: The first place can not indicated the abbreviation AF and BF, should be showed full name.

The authors thank the Reviewer for the correction. L. 27

The line 67-71, I suggest that rewrite the aim. Please reference some published papers.

The authors thank the reviewer for the suggestion and have rewritten the aim to make it clearer. L. 24-26; 67-77

I don’t know the meaning of the a b c A B C in the table1, 2. Such as 401 b B. Please show the meaning.

The authors apologize if the table captions were not clear. They have rewritten them trying to make them clearer. L. 210-215

The conclusion should add some future works for this study.

The conclusions were rewritten according to the suggestions of the other Reviewers. In the new version of the manuscript the authors have added the last sentence, reporting future works. L. 327-339

Please check the references format, carefully.

The authors thank the Reviewer for the suggestion.

Reviewer 3 Report

The manuscript titled "Do Wildfires cause changes in soil quality in short term?" is an interesting work and I recommend it for publication. However, I have some tips that could improve the work.

I recommend improving the abstract by adding some results and avoiding writing the list of analyses performed (line 22-23). Moreover, the opening sentence is too general and unclear (line 19-20). I suggest you to change it with a more specific but synthetic and exhaustive phrase. The same goes for keywords. The words are not representative of the work done. I would recommend more appropriate keywords such as sequential extraction instead of element availability and residual fraction. I would also underline the sampling area; a keyword could be Vesuvius National Park. Moreover, the word heavy metal is obsolete, could replace it with potential toxic trace element (PTTE). Finally, I don't think the word nutrient is an appropriate keyword for the paper. I think it is better to change it with a keyword that refers to the chemical properties analysed.

As regards the manuscript, it is very interesting and well written. However, I strongly recommend to improve the introduction section. In particular, the novelty of this article is expected to be improved. The opening phrase is unclear. What does “flammable plant species” mean? What does “among other causes” mean? Please can you explain these sentences. Moreover, line 42-43 are confused. Please justify and discuss them because the sentence seems incomplete. Finally, I recommend giving more emphasis to the aim of the work and improve this part. As concern Materials and Methods are clear and exhaustive. I only ask for a clarification: at line 126 the author write physical-chemical and biological parameters. It is possible add some physical parameters like soil texture for example. I think it can be an important information to value the chemical changes in burnt and unburnt soils. Results and discussion are well described and of adequate length. I have no comment on these parts of the manuscript. Conclusions are in line with previous results, however I recommend to write the conclusions differently. Indeed, Lines 290-300 seem to repeat the results and do not bring any news or conclusions. Moreover, I would advise to elaborate on the last paragraph (line 301-303), as it represents the novelty and possible developments of the work.

Author Response

Reviewer 3

The manuscript titled "Do Wildfires cause changes in soil quality in short term?" is an interesting work and I recommend it for publication. However, I have some tips that could improve the work.

The authors thank the Reviewer for the appreciation of the research. They also thank the Reviewer for the suggestions and corrections that improved the manuscript.

I recommend improving the abstract by adding some results and avoiding writing the list of analyses performed (line 22-23). Moreover, the opening sentence is too general and unclear (line 19-20). I suggest you to change it with a more specific but synthetic and exhaustive phrase.

The authors thank the Reviewer for the suggestion and have modified the abstract in the new version of the manuscript. L 21-23; 21-35

The same goes for keywords. The words are not representative of the work done. I would recommend more appropriate keywords such as sequential extraction instead of element availability and residual fraction. I would also underline the sampling area; a keyword could be Vesuvius National Park. Moreover, the word heavy metal is obsolete, could replace it with potential toxic trace element (PTTE). Finally, I don't think the word nutrient is an appropriate keyword for the paper. I think it is better to change it with a keyword that refers to the chemical properties analysed.

The authors thank the Reviewer for the correction and have replaced some key words. L. 37-38

As regards the manuscript, it is very interesting and well written. However, I strongly recommend to improve the introduction section. In particular, the novelty of this article is expected to be improved.

The opening phrase is unclear. What does “flammable plant species” mean? What does “among other causes” mean? Please can you explain these sentences.

Moreover, line 42-43 are confused. Please justify and discuss them because the sentence seems incomplete.

Finally, I recommend giving more emphasis to the aim of the work and improve this part.

The authors thank the reviewer for the suggestion. They have modified some parts of the introduction section and have stressed the novelty of the present research. L. 41-41; 46-47; 67-77

As concern Materials and Methods are clear and exhaustive. I only ask for a clarification: at line 126 the author write physical-chemical and biological parameters. It is possible add some physical parameters like soil texture for example. I think it can be an important information to value the chemical changes in burnt and unburnt soils.

The authors evaluated the texture in unburnt (silty-clay soils) but, unfortunately, they could not do that in burnt soils. Therefore, the authors have modified the title of the paragraph in the new version of the manuscript. L. 90-91; 111

Results and discussion are well described and of adequate length. I have no comment on these parts of the manuscript.

The authors thank the Reviewer for the attention deserved to this part of the manuscript.

Conclusions are in line with previous results, however I recommend to write the conclusions differently. Indeed, Lines 290-300 seem to repeat the results and do not bring any news or conclusions.

Moreover, I would advise to elaborate on the last paragraph (line 301-303), as it represents the novelty and possible developments of the work.

The authors completely rewrote the conclusions, avoiding reporting specific results, highlighting the novelty of the obtained results and adding perspectives for future works. L.327-339

Reviewer 4 Report

The work relates to monitoring research and has a local coverage, so interest in it will be limited. Nevertheless, it undertakes interesting issues, although presented in imperfect way. Study potential was not utilized due to inaccurate description of results and accidental, speculative discussion. There are many factual errors at work. All detailed information is entered directly in the manuscript. The work requires thorough rewording and even rewriting.

Author Response

Reviewer 4

The work relates to monitoring research and has a local coverage, so interest in it will be limited. Nevertheless, it undertakes interesting issues, although presented in imperfect way. Study potential was not utilized due to inaccurate description of results and accidental, speculative discussion. There are many factual errors at work. All detailed information is entered directly in the manuscript. The work requires thorough rewording and even rewriting.

The authors thank the Reviewer for the suggestions and corrections that improved the manuscript.

Abstract:

L.20 unclear, please to rewrite and improve professional English

L.23-24 unclear, please to rewrite and improve professional English

L. 23 bioavailability or availability ? - it's not the same and there is no word about bioavailability at paper

L.26- which C ? several forms of this element have been identified in the paper

L.29- how many soils were there? or maybe it was one type of soil and several samples. This should be written precisely

L.31- unclear, please improve the professional English

Keywords: 1 e 2 either one or the other 3: residual fraction or maybe: sequential extraction ?

Done. L. 21.23

Done. L. 26-27

The authors thank the Reviewer for the question. They have measured the element availability. Therefore, the authors have deleted this sentence.

The authors have measured total C and have specified that throughout the manuscript. L. 28

The authors have added the lacking information in the new version of the manuscript. L. 27

The authors have clarified the meaning of the sentence. L. 31-33

The authors have changed some keywords also according to the suggestion of another Reviewer. L. 37-38

Introduction:

L. 46-47- vague sentence. please write clearly

L.52- a very general statement and we do not know what the authors mean. please develop this thought by specifying and quoting the author

L.67 – factors not drivers

L.66-67 - please to rewrite

L.70-71 this sentence does not contain the research hypothesis, but the authors' expectations as to the results obtained. one should correctly formulate the research hypothesis giving tasks

The authors have modified the sentence. L. 49-52

The authors have modified the sentence. L. 56-58

The authors have deleted this part and they have rewritten the aim. L. 67-77

The authors have rewritten the sentence. L. 67-77

The authors thank the Reviewer for the suggestion and have added the hypothesis, of the research. L. 73-77

Materials and Methods

L.79-80 please to give the Latin systematic names for plants

L.87 basic information in this type of research is the granulometric composition of the soil samples tested and the soil systematics. This information must be completed !

L.89-90 meteorological conditions should be in graphical form for the whole year of research. may be included as supplementarny materials

L.90 a renewed doubt about the number and types of soils. or maybe the authors have in mind the number of soil samples from the same soil type.

L.91 at what distance from one site to another soil samples were taken, from what area? The whole park area? Parts of it? it should be described precisely. A map with marked soil sampling locations is also required and required.

L.91 AFTER LITTER REMOVAL please explain why this was done. What is this dictated?

L.92 why such a layer? there are also microbiological, biochemical and chemical transformations below? How and what soil samples were taken?

L.101 I understand that the authors refer to a literature source, but the presentation of results in professional scientific research should be based on real analysis and not conversion factors. The amount of OM from Corg can be converted for approximate purposes in monitoring studies. here a simple and short analysis should be performed, which will give reliable values.

L.106 Strange element selection, very random. please explain well and justify in the introduction why such elements were found in the work and important macronutrients such as P or S, essential microelements such as Zn or Ni were omitted

L.109 PSEUDO-TOTAL this approach is a serious factual error! The authors of the work only summed up the amounts of elements obtained in individual fractions. This cannot be equated with pseudo - total content, which is performed with aqua regia according to the ISO standard!

L.122 ML-1 superscript

L.129-132 completely obscure thought. Editorial required

L.131 MN completely obscure thought. Editorial require

L.142 DRIVERS maybe factors ??

L.156-158 a long sentence. illegible. reword

Done. L. 86-89

The authors evaluated the texture in unburnt but unfortunately they could not do that in burnt soils. So, the authors have cited a reference of a previous research. L. 90-91

In addition, they have added in the new version of the manuscript information regarding soil classification. L. 91

The authors have added a figure as required with meteorological data for the period September 2016 - September 2018. Figure A1

The authors have added more details about soil sampling, as required. In synthesis, the authors have collected soils at 12 field points inside the Vesuvius National park. L. 100-103

To be clearer, the authors have added a figure for the localization of the sampling field points and a spatial scale useful to identify the distance among them (Fig. 1).

The authors have removed the litter before sampling soils, as they were interested in soil characteristics. Besides, at the different field points the amount of remaining litter changed, making them not comparable.

The depth of the soils inside the Vesuvius National park varies according to the age (time of the material deposition deriving by the various volcanic eruptions) of the substrates from which the soils derive. Therefore, the authors sampled surface soils (0-10 cm), also to allow the comparison with the results of the campaign of 2016 (before fire occurrence). Finally, the authors use to sample the surface soil layer where the biological activity is higher (Ge et al., 2009, Plant and Soil 326: 31-44).

The authors thank the Reviewer for this comment and for giving the opportunity to clarify this point. It is widely used the evaluation of the organic matter by a conversion factor (Kimble and Follett, 2001, Assessment Methods for Soil Carbon, B.A. Stewart Ed.; Sleutel et al., 2007, Communications in Soil Science and Plant Analysis, 38:19-20; Pribyl, 2010, Geoderma, 156: 73-86), especially when the determination of the organic carbon content is performed by combustion. The authors have performed the determination of total and organic carbon through a N, C, S analyzer that completely burns the soil sample at high temperature. The authors preferred to perform the analyses of total and organic carbon through the same equipment in order to better compare them. In fact, the evaluation of soil organic matter through ignition would have been less useful for this aim.

In the present research, the authors aimed to evaluate the effects of fire on soil properties. To achieve the aim, they compared the results with those obtained in a campaign of 2016 (before fire), when the required elements were not measured. Anyway, in the present research, nutrients and potential toxic trace elements are well represented and useful to achieve the aim of the research. Anyway, in further investigations, that will aim to evaluate long-term effects, the precious suggestion of the Reviewer will be taken into account.

The pseudo-total approach is widely used in the scientific literature when element sequential extraction is performed. In previous research, the authors have compared the pseudo-total with the results deriving by the extraction according to ISO standard and narrow differences were detected between the two approaches. The first three sequential extractions are responsible of the different main soil availability, whereas the fourth one represents the residual fraction (the not available one). So, considering that the two approaches are technically very different and the final results (pseudo-total and ISO) are not so dissimilar, it could be useful save time and consumables when the researches perform the element sequential extraction to study the different availabilities. Finally, in the present research, the authors scarcely discuss the pseudo-total content, but are mainly interested in element availability. The researches are already published data using the pseudo-total (Memoli et al., 2018, Science of the Total Environment 625, 16–26) citation in the new version of the manuscript. L. 124

The authors thank the Reviewer for the correction. L. 139

The authors have used the Soil Quality Index (SQI) as it is widely used to estimate soil quality. The SQI is not restrictive and widespread and assumes the concept “more is better” or “less is better” functions. The authors have cited the authors that have proposed the SQI and the functions. L. 147-152

The authors have replaced “drivers” with “factors”. L. 161

The authors have rewritten the sentence. L. 175-177

Results

L. 164 discussion of the data contained in the tables and figures very general, casual. Important information omitted. This chapter must be rewritten and thoroughly discuss the data presented

L.166 -BF AND AF SOILS ??? soil or soil sampes

TABLE 1 the table lacks: C and N values ​​and EC

BF values in TABLE 1 Since when do we have more C than OM ???? Assuming the counting method given in line 101

L.174 concentration is determined in solutions! here we are talking about the content of elements in different connections with the solid phase of the soil described in fractions and expressed in mg / kg

L.74-76-77-78-79 SEE previous correction on BF AND AF SOILS and on PSEUDO-TOTAL CONCENTRTIONS

TABLE 2 INSTEAD OF SITES element

PSEUDOTOTAL Sum of fractions

I calculated all the values ​​in the whole table and in many places the sum does not match those given in the table. please verify it !!!

L.188-192 -1 superscript

L.199 C- which form C we are talking about

L.208 the values of SQI ..... amounted - please do not use scientific jargon

L.209 drivers it’s better factors

L.215 AF soils see previous correction

The authors have modified many parts of the discussion, providing more details and comments to better discuss the results.

The authors thank the Reviewer for the correction. They have clarified the experimental design in Material and Method section and replaced “soils” with “soil” throughout the new version of the manuscript.

The authors have reported in the table 1 the total C and N contents as reported in the caption. As already specified, some soil properties were not measured in soils after fire occurrence as they were not measured in the soil campaign before fire occurrence.

The authors have made a mistake and have corrected it in the new version of the manuscript. The authors thank the Reviewer to have highlighted the error.

The authors know that the element fractions are linked to a solid phase of the soil, but as they referred this content to soil dry weight, they expressed it as concentration. In Adamo et al., 2014 (Science of the Total Environment 500–501, 11–22), that performed the sequential extraction, the different fractions are also reported as concentrations. Anyway, according to your suggestion, the authors have replaced “concentrations” with “contents”.

Done.

The authors have corrected the error regarding the name of the column. The values reported in the “pseudo-total column” are the mean values of the pseudo-total contents calculated for each field points. Therefore, it cannot correspond to the sum of the four fractions, that are also the mean values among the 12 field points.

Done. L. 222-223

The authors have added the information throughout the manuscript. L. 235

The authors have corrected the sentence. L. 245

Done. L. 246

Done.

Discussion

General comment :

this chapter is based on speculations not supported by research. The authors focused only on selected results without interpreting them. Lack of discussion to changes in the quantities of elements in fractions II and IV (the latter are devoted to conclusions in the absence of its discussion!). Chapter for thorough rewording.

Please explain the observed increase or decrease of element contents.  It is understandable that during a fire there is a loss of OM and water evaporation, but the elements remain in the soil - they have not evaporated, because they are not in gaseous forms! So how do you explain their loss? certainly make other connections with the steel phase of the soil here mainly with the mineral part - and this aspect should be discussed.

L.218 not all elements are nutrients ! for example: Al, Cr, V, Pb

L.219 what C, we are talking about content no concentrations !

L. 221 what faction is that?

L.222 speculation! no such studies were carried out at work

L. 224speculation! no such studies were carried out at work

L.226 ACCORDING TO SEVERAL STUDIES please to explain it

L.235 PSEUDO-TOTAL AND ELEMENT CONCENTRATIONS see previous suggestions

L.239 And what about Mn- his quantity also increased, although it was not a significant change, it should also be noted

L.241 this is not true in Fr. I the amounts of 5 elements increased and 5 decreased - what does this mean?

L.242 AF SOILS AND BF SOILS see previous suggestions

L.246 this aspect was not analyzed, so they are speculations

L.252-253 why there is such a dependence - please explain

L.252 ELEMENT AIVABILITY how to explain the tendency to increase in quantity of elements in Fr.II and Fr. IV

L.254 e L.261 AF SOILS AND BF SOILS see previous suggestions

L.264 There is no data concernig to soil oragnic matter fractionation ! therefore, you cannot compare to lietratura data in this regard

L.265 What about Fe, Pb and V? - their quantities also increased as shown in Table 2. Why did they not affect DHA?

L.266 F2 and F3 ?

L.267-289 please to edit by specifying which particular elements studied are mentioned

The authors thank the reviewer for the suggestions that have contributed to improve the discussion.

The authors have provided more comments about F2 and F4. L. 280-290

In addition, they also improved the discussion of soil changes due to fire. L..

Regarding the observed decrease of some elements this can be due to erosion or leaching phenomena. In fact, after organic matter destruction, a great amount of elements can be solubilized in soil water and then reach the deeper layers. In addition, some elements can be also volatilized due to suspension of the fine soil particles by the wind.

The authors thank the Reviewer for the correction. L. 256

The authors have specified that C is total C content. L. 257

This part of the discussion is not referred to a particular available fraction. In fact, the authors referred to the belonging of carbon compounds characterized by different flammability. In particular, the observed decrease of carbon could be due to high inflammable organic compounds; whereas, the carbon that remains in the soil likely belongs to scarce flammable organic compounds or compounds protected by soil components. In order to better explain these concepts, the authors have rewritten the sentence. L. 257-260

The authors agree with the Reviewer suggestion and have deleted the sentence.

The authors have modified the sentence. L. 260-263

Done.

Besides, they have corrected the error throughout the new version of the manuscript.

The authors thank the Reviewer for the comment and they focused the discussion only on the results that showed statistically differences. Anyway, according to the request, the authors have added discussion on Mn trend. L. 274-276

The authors thank the Reviewer for the correction. L 276-277

Done.

The authors have deleted the sentence, according to the Reviewer request.

The authors changed the sentence to explain the dependence of microbial community to soil abiotic properties. L. 291-293

The authors have added this information in the new version of the manuscript. L. 280-290

Done.

The authors have deleted this part in the new version of the manuscript.

Multiple Regressions are statistical models, searching a significant effect of independent variables (in this case element and soil properties) on one dependent variable (in this case each biological one), identifying the most statistically impacting independent variables. In the present research they were Al, Cr and Cu contents.

The authors have made a mistake and have corrected it. L. 308

The authors have changed the sentence. L. 318-321

Conclusions

General comment: these are not conclusions, only repeated results. There are references to data not presented and not discussed at work!

L.290 C CONCENTRATIONS see previous suggestions

L.292 PERCENTAGE this work was not presented!

L.293 not only, see: Pb, Cu, Al !

L.294-300 repeated results

The authors thank the Reviewer for the suggestions and they have changed the conclusions to avoid repetitions.

Done.

The authors have made a mistake and have deleted this part.

The authors have modified the conclusions deleting this part.

The authors have avoided repetitions in the new version of the manuscript.

Round 2

Reviewer 4 Report

I would like to thank the authors for responding to my comments and including most of them. The new version of the work is significantly improved and the reception is now much better. Nevertheless, I still have some suggestions for fragments of the work. The explanations that I got regarding the removal of plant material and the depth of sampling from 0-10 cm should also be included at paper, so as to fully reflect the accuracy of the studies carried out. Moreover, I still find the description for Table 2 too laconic - 2 sentences to very little with so much data. Similary the part of  discussion devoted to this, it should be expanded. I understand the authors' explanations related to the calculations for the amount of "pseudo-total", but I still disagree with this, the more so that the description of the table clearly shows the summation of the average values ​​of the elements in individual fractions. Therefore the averaged values ​​of individual fractions for individual soils and the elements in the table should be equal to the given values.Otherwise, the reader is misled. The more so that this method of data presentation is indicated in the methodology, quoting the literature. Moreover, in table 2 in column 2 I propose to replace the heading "fire" to "soil" or "site"

Author Response

I would like to thank the authors for responding to my comments and including most of them. The new version of the work is significantly improved and the reception is now much better. Nevertheless, I still have some suggestions for fragments of the work.

The authors thank the Reviewer for the further suggestions and corrections that improved the manuscript.

The explanations that I got regarding the removal of plant material and the depth of sampling from 0-10 cm should also be included at paper, so as to fully reflect the accuracy of the studies carried out.

Done. L. 102-104

Moreover, I still find the description for Table 2 too laconic - 2 sentences to very little with so much data.

Similary the part of discussion devoted to this, it should be expanded.

The authors have added more details in the result section and have also discussed them. L. 209-213; 285-289; 293, 295.

I understand the authors' explanations related to the calculations for the amount of "pseudo-total", but I still disagree with this, the more so that the description of the table clearly shows the summation of the average values ​​of the elements in individual fractions. Therefore, the averaged values ​​of individual fractions for individual soils and the elements in the table should be equal to the given values. Otherwise, the reader is misled.

The more so that this method of data presentation is indicated in the methodology, quoting the literature.

The authors have calculated the pseudo-total according to the suggestion of the Reviewer.

Moreover, in table 2 in column 2 I propose to replace the heading "fire" to "soil" or "site"

Done.